# Synthesis and Characterisation of Poly(3-hydroxybutyrate-*co*-3-hydroxyvalerate)-*b*-poly(3-hydroxybutyrate-*co*-3-hydroxyvalerate) Multi-Block Copolymers Produced Using Diisocyanate Chemistry

**DOI:** 10.3390/polym15153257

**Published:** 2023-07-31

**Authors:** Jingjing Mai, Steven Pratt, Bronwyn Laycock, Clement Matthew Chan

**Affiliations:** School of Chemical Engineering, The University of Queensland, Brisbane, QLD 4072, Australia; jingjing.mai@uq.edu.au (J.M.); s.pratt@uq.edu.au (S.P.)

**Keywords:** polyhydroxyalkanoates, isocyanate chemistry, block copolymers, allophanates, cross-linking

## Abstract

Bacterially derived polyhydroxyalkanoates (PHAs) are attractive alternatives to commodity petroleum-derived plastics. The most common forms of the short chain length (scl-) PHAs, including poly(3-hydroxybutyrate) (P3HB) and poly(3-hydroxybutyrate-*co*-3-hydroxyvalerate) (PHBV), are currently limited in application because they are relatively stiff and brittle. The synthesis of PHA-*b*-PHA block copolymers could enhance the physical properties of PHAs. Therefore, this work explores the synthesis of PHBV-*b*-PHBV using relatively high molecular weight hydroxy-functionalised PHBV starting materials, coupled using facile diisocyanate chemistry, delivering industrially relevant high-molecular-weight block copolymeric products. A two-step synthesis approach was compared with a one-step approach, both of which resulted in successful block copolymer production. However, the two-step synthesis was shown to be less effective in building molecular weight. Both synthetic approaches were affected by additional isocyanate reactions resulting in the formation of by-products such as allophanate and likely biuret groups, which delivered partial cross-linking and higher molecular weights in the resulting multi-block products, identified for the first time as likely and significant by-products in such reactions, affecting the product performance.

## 1. Introduction

Polyhydroxyalkanoates (PHAs) are a family of natural polyesters that are synthesised by microorganisms as an intracellular material to store carbon and energy [1]. Their biocompatibility and biodegradability make them excellent candidates for biomedical applications [2,3,4]. They are also thermoplastic and/or elastomeric (depending on their copolymer composition), which makes them attractive substitutes for petroleum-based synthetic polymers such as polypropylene (PP), polyethylene (PE) and polyethylene terephthalate (PET) [5,6].

However, the most common of the PHAs, the homopolymer poly(3-hydroxybutyrate) (P3HB) and copolymer poly(3-hydroxybutyrate-*co*-3-hydroxyvalerate) (PHBV), are constrained in their use by limitations in their material properties. P3HB and PHBV with low (1 mol%) 3-hydroxyvalerate (3HV) content, in particular, have very narrow processing windows and are relatively brittle [7,8]. The synthesis of block copolymers based on these starting materials as one block, but coupled to another block of PHA that is incompatible, is one approach for the potential step-change improvement in PHA properties [9,10,11,12]. 

The production of blocks of copolymers rather than blends is a highly attractive option because of their versatility in developing unique microstructures that deliver very different material properties for the as-formed products, as well as the flexibility to fabricate variations in polymer chain aggregation [8,9,11,13,14]. Furthermore, the molecular weight, to a large extent, is responsible for the end-use properties of biopolymers [15,16]. The mechanical properties of PHBV random copolymer, for example, deteriorate when the weight average molecular weight ($M_{w}$) is lower than 100 kDa [17,18]. So, the target of this work is to produce PHA-*b*-PHA block copolymers of high molecular weight ($M_{w}$> 100 kDa). 

Attempts to produce PHA-*b*-PHA block copolymers have previously been made using a biological approach. Two strategies have been adopted: one using direct pulse feeding of carbon feedstock to unmodified organisms and deliberately switching on and off the feeding over time and/or alternating the feedstock to produce different blocks [8,9]. It was concluded that such a biological approach lacks control over the timing and extent of block formation and in all probability produces both PHA/PHA blends as well as blocks and is thus a complex mixture for which structure–property relationships of PHA-*b*-PHA of different block lengths and compositions cannot readily be assessed [10,11,14,19,20,21,22,23]. The other strategy is to use genetically modified organisms and to manipulate the switching on and off of the metabolic pathways to deliver more controlled PHA-*b*-PHA blocks [10,11,22,23]. However, this strategy needs very tight control over the organisms present, with expensive processing and genetic manipulation, and also requires advanced biological manipulation of the synthetic strategy. 

By contrast, the chemical synthesis of P3HB-/PHBV-based block copolymers using natural random copolymers as starting materials is a reasonably straightforward synthetic process that has the potential to deliver relatively controllable material structures and narrowly defined compositions [24,25,26]. Various chemical methods can be applied for the synthesis of PHA-based block copolymers, such as living radical polymerization, click reaction, transesterification, isocyanate coupling reaction and ring opening polymerisation, with Samui and Kamai providing a recent review [27]. However, only limited chemical methods have been used to synthesise PHA-*b*-PHA block copolymers, due to the absence of functional groups such as vinyl groups in most of the natural PHAs [28].

Further, in general, the typical PHA-based block copolymer synthesis reported in the literature has used relatively low molecular weight hydroxy-functionalised PHA oligomers (i.e., $M_{n}$ < 4 kDa) derived from high molecular weight random copolymers as the starting materials. However, for some applications such as packaging, high molecular weight final products are needed, to achieve good mechanical properties. When starting with such low molecular weight materials, it is therefore necessary to produce multi-block copolymers in order to obtain relatively high molecular weight polymers with good mechanical properties [29,30,31]. However, this strategy is less controllable with respect to the chemical structure and architecture of the final products and introduces multiple urethane linkages, likely influencing the final product properties. In addition, one of the dominant side reactions is the formation of allophanates, which has been reported in polycaprolactone (PCL)-*b*-polyethylene glycol (PEG) and PHA-*b*-PEG systems [32,33]. This resulted in blocks with higher molecular weights and partially cross-linked products which could lead to a shift in properties and the formation of microgels [32,33]. Despite being reported, the allophanate side reactions were not characterised further in the past literature and none was reported in PHA-*b*-PHA systems.

Further, since much of the potential property improvement in block copolymers is obtained through local microphase separation of the different blocks [9,34,35,36], longer block sequences may be desirable. Yet, the one example to our knowledge of the synthesis of a PHA-*b*-PHA copolymer using the diisocyanate approach is where a low molecular weight block copoly(ester–urethane) of $M_{n}$ = 10.6 kDa was synthesised using a one-step process from telechelic hydroxylated poly[(*R*)-3-hydroxyoctanoate] (PHO-diol, $M_{n}$ = 2.4 kDa) and telechelic hydroxylated poly[(*R*)-3-hydroxybutyrate] (P3HB-diol, $M_{n}$ = 2.6 kDa) with L-lysine methyl ester diisocyanate (LDI) as the junction unit. The synthesised block copolymer delivered good thermoplastic properties, with a melting temperature (*T_m_*) of 146 °C and a glass transition temperature (*T_g_*) of −6 °C. The mechanical properties were however those of a soft, low-strength material, consistent with its low molecular weight [37].

Therefore, the aim of this work was to synthesise novel, high molecular-weight PHBV-*b*-PHBV block copolymers, using PHA macroinitiators of relatively high molecular weights (e.g.,$\;M_{n}$ > 20 kDa).

The main approach to the chemical synthesis of PHA-based block copolymers is the use of isocyanate chemistry to form urethane linkages from hydroxy-terminated PHA blocks. This is a simple and efficient synthesis strategy, being quite rapid and clean, and being a well-established process [38]. PHA-based block copolymers can be produced using either a one-step or two-step process, where a one-step process adopts a single synthesis stage, with all reactants combined together at the start, while a two-step process adopts the alternate strategy of reacting a central hydroxy-terminated block with at least two times the molar equivalent of diisocyanate to produce a central isocyanate-terminated block, which is then subsequently reacted with an alternate hydroxy-terminated block in a second stage. There are now many examples of these strategies being adopted in the literature, leading to a wide range of block copolymeric products, often of significant molecular weight, with distinctly different properties to their starting materials [4,24,34,37,39,40,41]. Overall, PHA-based block copolymers joined by urethane links typically exhibit high thermal and mechanical properties and good processing ability over their counterpart random and blend copolymers [34,42]. However, this strategy has not been applied to PHBV-*b*-PHBV synthesis, and it is a particularly relevant one for targeting high molecular weight block copolymers where the proportion of functional end groups relative to the length of the main chain is low, needing efficient coupling.

In this first instance, the methods for synthesis using isocyanate chemistry were established using 1 mol% 3HV blocks. The effectiveness of both the one-step and two-step processes for the production of PHBV-*b*-PHBV multi-block copolymers were compared by tracing the reaction chemistry and the molecular weight of the products throughout. The extent of possible side reactions, i.e., the formation of allophanates, was also characterised. Our findings provide some insight into the methodology for the synthetic production of PHA-*b*-PHA block copolymers, resulting in block copolymers of industrially relevant molecular weight. 

## 2. Materials and Methods

### 2.1. Materials

A commercial PHBV random copolymer of 1 mol% 3HV content was purchased from TianAn Biopolymer (Ningbo, China) ($M_{n}$ = 194 kg/mol, $M_{w}$ = 455 kg/mol and *Đ* = 2.3, by GPC, 1 mol% 3HV, by ^1^H-NMR, called Random_1HV#3 in this work). Hydroxy-functionalised PHBV copolymers of low (1 mol%) 3HV content (Random_1HV#1, Random_1HV#2, Random_1HV#4 and Random_1HV#5) were produced, as described in our previous work [43]. Details of these materials are provided in the Appendix A, noting that these products are mixtures containing mono-hydroxy terminated PHBV and di-hydroxy-terminated (telechelic) PHBV, with some carboxylic acid group functionality as well. HPLC grade chloroform, 1,2-dichloroethane (anhydrous, 99.8%), tin(II) 2-ethylhexanoate (stannous octoate, 92.5–100.0%) and hexamethylene diisocyanate (puriss., ≥99.0% (GC)) (HDI) were purchased from Sigma-Aldrich (St. Louis, MI, USA) and used as received. Deuterated chloroform (99.8%) was purchased from Novachem (Calgary, AB, Canada) and used as received. Argon (Ultra High Purity) was purchased from Supagas Pty Ltd. (Branxholm, Australia) and used as received.

### 2.2. Block Copolymer Synthesis

Based on our previous work and the literature [4,44,45], three different experiments using hydroxy-terminated PHBV macromer starting materials and either a one-step or a two-step process were carried out. The experimental conditions, including the details of the hydroxy-terminated PHBV macromer used in these experiments and whether or not the experiment was multi-step, are shown in Table 1. A summary of the experimental approach is provided in Figure 1. Experiment #1 was a two-step synthesis. In the first step, the hydroxy-functionalised PHBV (Random_1HV#4) solution was added dropwise to hexamethylene diisocyanate (HDI) at an [NCO]/[OH] ratio of 2.2 and temperature of 65 °C. The reaction of the first step lasted for 5 h. In the second step, double portions of hydroxy-functionalised PHBV (Random_1HV#4) solution were quickly added into the reaction system and the reaction was run for 48 h. Only the reaction products in the second step of the two-step synthesis were analysed.

Experiment 2 was a one-step synthesis. Experiment #2A was essentially a repeat of the first step of Experiment #1, and so the first 5 hours of it were used to understand the reaction kinetics of the first step in a two-step synthesis. Experiment #2B was the duplicate experiment of Experiment #2A with extended time and samples taken throughout so as to follow the reaction kinetics of the one-step synthesis.

#### 2.2.1. Drying of Glassware, Solvents and PHBV Reagents

In all these experiments, glassware was dried in an oven at 200 °C overnight and then cooled in a desiccator over dried silica gel before use. The solvent 1,2-dichloroethane was dried before use [46] and distilled by azeotropic distillation using a short-path distillation apparatus. In this process, a two-necked round-bottomed flask (250 mL) equipped with a reflux condenser and gas inlet was connected to a bubbler and the glassware was flushed with argon and carefully flame dried. Then, 100 mL 1,2-dichloroethane and 3 g calcium hydride were quickly added into the flask, which was heated in an oil bath at 60 °C under an argon flux and magnetically stirred for 2 h. The reflux condenser was then replaced with a short-path distillation head, which was connected to a one-necked receiving flask (100 mL) and a bubbler. The pre-dried 1,2-dichloroethane was then dried further by azeotropic distillation, with the first 10% of azeotrope collected in the receiving flask being discarded to remove the trace water from the solvent. The remaining distillate was collected and the flask was flushed with argon before being sealed with a rubber septum and parafilm as a dried solvent for the subsequent isocyanate reactions.

The hydroxy-terminated PHBV was firstly dried under vacuum (−90 kPa) in an oven at 80 °C overnight then cooled in a desiccator over dried silica gel before use. A two-necked round-bottomed flask (250 mL) equipped with a rubber septum (sealed with parafilm) and a reflux condenser was connected to a bubbler and the glassware was flushed with argon by injecting a needle through the rubber septum. The glassware was carefully flame dried before dried hydroxy-terminated PHBV was quickly added into the flask under an argon stream. Then, dried 1,2-dichloroethene (5 mL/gPHA) was quickly added into the flask through the rubber septum using a 50 mL glass syringe, again under an argon stream. The flask was then heated in an oil bath at 140 °C under an argon flux and magnetically stirred until the PHBV was fully dissolved. Then, the solution was dried by azeotropic distillation until 10% of the azeotrope was collected in the receiving flask (50 mL). The oil bath was then cooled down to 65 °C. 

#### 2.2.2. Experiment #1: Two-Step Synthesis of PHBV-*b*-PHBV Multi-Block Copolymer

A three-necked round-bottomed flask (100 mL) equipped with a reflux condenser and gas inlet was connected to a bubbler and the glassware was flushed with argon and carefully flame dried. In the first step of the block copolymer synthesis, 3 g dried hydroxy-terminated PHBV random copolymer (Random_1HV#4) was dissolved in 15 mL dried 1,2-dichloroethane and the PHBV solution was dried by azeotropic distillation. After the PHBV solution was prepared, the reaction apparatus was flame dried again. Then, 2 mL 1,2-dichloroethane, 0.012 mL stannous octoate as a catalyst (0.8 µL per mL 1,2-dichloroethene) and 0.044 mL hexamethylene diisocyanate (HDI) (based on desired ratio relative to the concentration of OH groups) were quickly added into the flask through the rubber septum using 500 µL glass syringes. The solution was magnetically stirred and heated in an oil bath at 65 °C. Then, the PHBV solution was immediately transferred into the three-necked flask dropwise through the rubber septum using 20 mL glass syringes. The reaction was conducted for 5 h. 

In the second step, 6 g of the hydroxy-functionalised PHBV random copolymer was dissolved in 30 mL dried 1,2-dichloroethane and the PHBV solution was dried by azeotropic distillation. Then, the dried PHBV solution was quickly transferred into the three-necked flask through the rubber septum using 20 mL glass syringes. The reaction was conducted for 48 h. In this second step, the [NCO]/[OH] ratio was 0.5, in accordance with normal practice, in order to leave hydroxyl end groups present post synthesis. Samples were collected by glass syringes through the rubber septum at different reaction times and quenched in a mixture of diethyl ether and methanol (20/1, *v*/*v*) for further characterisation. Finally, the quenched solids were recovered by vacuum filtration through a Buchner funnel fitted with a dried, pre-weighed quantitative filter paper (Whatman^TM^, Maidstone, UK, 10312209) and rinsed with a large volume of methanol. Then, the product was dried in a vacuum oven with a negative pressure of −90 kPa at 60 °C overnight.

#### 2.2.3. Experiment #2A: One-Step Synthesis of PHBV-*b*-PHBV Multi-Block Copolymers

A three-necked round-bottomed flask (100 mL) equipped with a reflux condenser and gas inlet was connected to a bubbler and the glassware was flushed with argon and carefully flame dried. Then, 5 mL 1,2-dichloroethane, 0.032 mL stannous octoate as a catalyst (0.8 µL per mL 1,2-dichloroethene) and 0.118 mL hexamethylene diisocyanate (HDI) (based on desired ratio relative to concentration of OH groups) were quickly added into the flask, through the rubber septum, using 5 mL and 500 µL glass syringes, respectively. The solution was magnetically stirred and heated in an oil bath at 65 °C. Then, the dried hydroxy-functionalised PHBV solution (containing 8 g PHBV) was immediately transferred into the three-necked flask dropwise through the rubber septum using a 20 mL glass syringe.

The reaction was conducted for 18.5 h. Samples were collected by glass syringes through the rubber septum at different reaction times and quenched in a mixture of diethyl ether and methanol (20/1, *v*/*v*) for further characterisation. Finally, the quenched solids were recovered by vacuum filtration through a Buchner funnel fitted with a dried, pre-weighed quantitative filter paper (Whatman^TM^, Maidstone, UK, 10312209) and rinsed with a large volume of methanol. Then, the product was dried in a vacuum oven with a negative pressure of −90 kPa at 60 °C overnight.

#### 2.2.4. Experiment #2B: One-Step Synthesis of PHBV-*b*-PHBV Multi-Block Copolymers with Longer Reaction Time

Experiment #2B was a duplicate experiment of Experiment #2A, using a hydroxy-functionalised PHBV (Random_1HV#5) of the same 3HV content and similar molecular weight, with the reaction being run for a longer time (24 h). In this experiment, 8 g hydroxy-terminated PHBV (Random_1HV#5) was reacted with 0.145 mL hexamethylene diisocyanate (HDI) (based on the desired NCO ratio relative to concentration of OH groups, see Table 1). The protocol was otherwise as described in Section 2.2.3.

### 2.3. Preparation of PHBV Films by Solvent Casting

To prepare solvent cast films, 1 g PHBV was dissolved in 10 mL HPLC chloroform (100 g/L) at 80 °C in an oil bath under reflux and cooled down after being well dissolved. Then, the solution was placed in a Petri dish. After most of the solvent was slowly evaporated, the film was dried under a vacuum to constant weight at room temperature. Then, the film was left in an open environment for aging for at least two weeks before testing.

### 2.4. Solubility Assessment for As-Produced Block Copolymeric Products

The as-produced products were dissolved in HPLC-grade chloroform (25 mg/mL) at 80 °C in an oil bath under reflux. After the PHBV was well dissolved, the solution was cooled and filtered through a Buchner funnel using pre-weighed polytetrafluoroethylene filter papers (0.22 µm, Whatman^TM^, Maidstone, UK, WM4-022090). Then, the filtered solution was transferred into pre-weighed Petri dishes followed by rinsing of the receiving flask three times with 20 mL chloroform each time, which was then added to the petri dish. Then, the filtered solution was allowed to evaporate in the fume hood. The mass of the solid residue was measured. The filter paper and the remaining undissolved solid were transferred into a separate Petri dish and the mass of the remaining solid was weighed after evaporation of the solvent (when it had reached constant weight).

### 2.5. Material Characterization

#### 2.5.1. Gel Permeation Chromatography (GPC)

The number average molar mass ($M_{n}$), weight average molar mass ($M_{w}$) and dispersity (*Đ*) of the PHBV copolymers were determined by GPC, using an Agilent 1260 Infinity Multi Detector Suite system (Cheshire, UK). Samples were dissolved in HPLC grade chloroform (2.5 mg/mL) followed by filtration using polytetrafluoroethylene syringe filters (0.22 µm, Kinesis, ESF-PT-13-022). An HPLC solvent delivery system was used in conjunction with an auto-injector. A column set consisting of a guard column (Agilent PLgel 10 µm 7.5 mm × 50 mm) followed 3 × Agilent PLgel 10 µm MIXED-B columns (7.5 mm × 300 mm) in series. The columns were kept at 30 °C. A refractometer, at 30 °C, was used to detect the signals. A chloroform flow rate of 1 mL/min was used for the analysis. Narrowly distributed molecular weight polystyrene standards and Agilent PS-H EasiVial calibration standards (PL2010-0201). The apparatus was calibrated with polystyrene standards. The Mark–Houwink–Sakurada (MHS) relation and specific MHS parameters (*K* and *a*) were used to correct the molar mass. $K=7.7\times 10^{-3}$ mL/g and α=0.82 were used in this study [47].

#### 2.5.2. Nuclear Magnetic Resonance Spectroscopy (NMR)

Quantitative ^1^H high-resolution one-dimensional NMR spectra were acquired at 298 K in deuterated chloroform (CDCl_3_) (50 mg/mL) on Bruker Advance 700 and 500 spectrometers. The number of scans was set to 512. The relative peak intensities of ^1^H-NMR spectra were determined using PeakFit 4.12 software [9]. Chemical shifts were referenced to the residual proton peak of CDCl_3_ at 7.26 ppm. The list of chemical shifts of the as-identified peaks from ^1^H-NMR throughout the study are presented in Appendix A.

Solid-state ^13^C-NMR spectra were performed at 298 K on a Bruker Advance III spectrometer with a 300 MHz magnet equipped with a 4 mm double air bearing, magic angle spinning probe. The powdered samples were placed in a zirconia rotor with a Kel-F cap and rotated at 5 kHz. Next, ^13^C spectra were recorded with a CPMAS pulse sequence. The ramped cross-polarization time was 1 ms, and decoupling was carried out using a tppm 15 sequence with 100 kHz. The relaxation delay was 3 s and the acquisition time was 49 ms. A total of 2048 scans were collected. Adamantane was used as a reference. The list of chemical shifts of the as-identified peaks from ^13^C-NMR throughout the study are presented in Appendix A.

#### 2.5.3. Differential Scanning Calorimetry (DSC)

DSC analysis was performed using a TA instrument Q2000. Samples of 2–4 mg in sealed aluminium pans were analysed under nitrogen flow (50 mL/min). A five-step procedure was applied as follows: (1) Equilibrate at 25 °C, then heat up from 25 °C to 190 °C with a 10 °C/min ramp and keep isothermal for 0.1 min to erase the thermal history; (2) cool down to −70 °C with a 10 °C/min ramp and keep isothermal for 5 min; (3) heat up from −70 °C to 190 °C with a 10 °C/min ramp; (4) cool down to −70 °C with a 100 °C/min ramp; and (5) heat up to 25 °C with a 20 °C/min ramp. The melting temperature ($T_{m}$) and enthalpy of melting ($∆H_{m}$) were determined from the first heating cycle while the crystallisation temperature ($T_{c}$) was determined from the first cooling scan. The glass transition temperature ($T_{g}$) was determined from the final heating cycle. Data were analysed using TA Universal Analysis software (UA 4.5.0.5).

## 3. Results and Discussion

In this work, the synthesis of PHBV-*b*-PHBV block copolymers was achieved using diisocyanate chemistry, adopting either a two-step or a one-step approach. The starting materials were relatively high molecular weight hydroxy-terminated PHBV macromers of (1 mol%) 3HV content, and the diisocyanate was deliberately selected to be the less rigid aliphatic hexamethylene diisocyanate (HDI), to limit the risk of introducing rigidity into the resulting chain.

The loading of isocyanate to the reactive end group is an important variable. Excess of isocyanates, over the stoichiometric requirements, i.e., [NCO]/[OH] ratio of 2.2, was used in Experiment 2A. 

### 3.1. Molecular Weight and Functional Groups of the Reaction Products

An increase in molecular weight is a key indicator for the success of the block copolymer synthesis with results shown in Figure 2. It should be noted that further reaction of the isocyanate with the urethane reaction products to produce allophanate and/or biuret bonds is possible, causing branching and chemical cross-linking [48,49]. Therefore, it is likely that after all the hydroxyl groups were reacted with isocyanate groups (Figure 3a,b), the further increase in molecular weight was mainly dependent on the formation of allophanate linkages by the reaction of urethane groups with isocyanate groups (Figure 3c), which has been illustrated in our previous study [44].

The proportion of insoluble components in the final products was assessed as an indicator of the extent of cross-linking (Table 2). A gelatinous insoluble component was observed, at 2.7–19.5 wt.%, indicating some by-product formation. In addition, the one-step products (Block_1HV#1 and Block_1HV#2) were characterised using solid-state NMR and compared with their counterpart random copolymers of the same 3HV contents and similar molecular weights (Random_1HV#1 and Random_1HV#2). As shown in Figure 4, peaks at around 28 ppm were only observed in the one-step products, which were attributed to the methylene group of an allophanate functionality (labelled as Al_2_, 4C, -O-CO-(R)N-CO-NH-CH_2_-CH_2_-CH_2_-CH_2_-CH_2_-CH_2_-) [44,50,51], indicating the formation of allophanate groups (therefore cross-linking) in the block copolymer synthesis. This was also evidenced by the broad peaks at around 142–160 ppm, which were attributed to the carbonyl peaks of the allophanate (labelled as C_1_, 1C, -O-CO-(R)N-CO-NH-CH_2_-) [50,51]. The detailed chemical shifts of the other peaks are shown in the Appendix A. Thus, the gel as produced was likely primarily due to the formation of these allophanate cross-linking groups. 

#### 3.1.1. Two-Step Synthesis (Experiment #1) 

The molecular weight was quantified through only the final (second) step of the two-step reaction for Experiment #1 (shown in Figure 2a); the initial 5 h of reaction was inferred from the first 5 h of Experiment #2A. It is worth noting that the molecular weights of the mixture at time zero in the second step (i.e., after adding the second hydroxy-functionalised block, Random_1HV#4, $M_{n}$ of 28 kDa, $M_{w}$ of 51 kDa) are smaller than the product after 5 h of reaction for Experiment #2A ($M_{n}$ of 43 kDa, $M_{w}$ of 81 kDa). This is due to the fact that the addition of hydroxy-functionalised PHBV at the start of the second step decreased the average molecular weight of the mixture. Regardless, excess isocyanate was still expected to be present at the start of this second stage of the two-step block copolymer synthesis. This was evidently the case with the molecular weight of the reaction products in the second step increasing gradually, with $M_{n}$ increasing from 28 to 43 kDa and $M_{w}$ increasing from 56 to 74 kDa within 3 h of the addition of the second aliquot of hydroxy-functionalised PHBV (Figure 2a).

The [NCO]/[OH] ratio over the whole synthesis was 2.2:3. But this meant that in the second stage of this two-step synthesis, the [NCO]/[OH] ratio would be at best less than 1, even if no reaction had occurred in the initial 5 h stage. If this reaction had run as planned, the molecular weight should in the end have been triple that of the time zero material. However, the molecular weight of the final reaction products only increased around 1.5 times after 3 h in this second stage. And after that, no further increase in molecular weights was observed even after an extended 48 h, i.e., the addition of the second dose of hydroxy-functionalised PHBV effectively consumed all remaining isocyanate but without further blocking copolymer synthesis. As shown in Figure 5b, the ^1^H-NMR spectra showed that neither the primary (Et_1_ and Et_2_) nor the secondary hydroxyl groups (2 °OH) were fully consumed after 48 h, while urethane peaks associated with the reaction of the primary and secondary hydroxyl groups (Ur_1_ (1H, -O-CH_2_-CH_2_-O-CO-NH-) and Ur_2_ (1H, -CO-CH_2_-CH(CH_3_)-O-CO-NH-), respectively) were observed in all reaction products [44], confirming that there was insufficient isocyanate present at the end to complete the reaction. There are many possible reasons for this, not least of which is that a large proportion of isocyanate has been consumed in the first step, with isocyanate reacting with both hydroxyl groups and urethane groups. It is also noted that trace water or other impurities present in the second batch of PHBV copolymers or introduced during their addition reacted with any unreacted diisocyanate or isocyanate functionalised PHBV that was present, removing the potential for further reaction. The reaction product was thus likely a blend of some block material with unreacted hydroxy-terminated PHBV macromonomer.

Although this experiment failed to build up molecular weight as expected, there was still 2.7 wt.% of the insoluble component in the final products, which is likely due to the formation of cross-linked materials caused by excess isocyanate in the first step.

#### 3.1.2. One-Step Synthesis (Experiment #2A and 2B)

In contrast to the two-step synthesis, the molecular weights through time of the reaction products from Experiment #2A showed a large increase between the 5 h reaction product and the 18.5 h reaction product, with the $M_{n}$ increasing from an initial 28 kDa to reach 93 kDa and $M_{w}$ increasing from 51 kDa to 163 kDa after 18.5 h (Figure 2b). Both the $M_{n}$ and the $M_{w}$ of the final reaction products were greater than triple that of the starting hydroxy-functionalised PHBV, indicating the formation of a multi-block copolymer. It is clear that under these conditions, and with the excess isocyanate present, the isocyanate reaction continued over time to produce a likely blend of different multi-block copolymers of PHBV, as would be expected.

The ^1^H-NMR spectra of the reaction products of Experiment #2A are shown in Figure 5a and revealed that the primary hydroxyl groups were fully consumed within 40 min, while the secondary hydroxyl groups were fully consumed within 1.5 h. The urethane peaks associated with the reaction of the primary and secondary hydroxyl groups were observed in all reaction products, confirming the proposed isocyanate chemistry. The presence of trace allophonate groups (labelled as Al_1_, 2H, -O-CO-(R)N-CO-NH-CH_2_-CH_2_-CH_2_-CH_2_-CH_2_-CH_2_-) was also confirmed [44], which was consistent with solid-state NMR results.

Overall, from the results of Experiment #1 and #2A, the idea of preparing PHBV capped with isocyanate groups at each end in the first step and then synthesising a tri-block copolymer was shown to be challenging under the experimental conditions used. This is different from what has been reported in the literature, where tri-block copolymers have been successfully synthesised using a two-step strategy at 50 °C with a [NCO]/[OH] ratio of 2.2 in the first step [45,52]. However, in our work, the formation of allophanate groups coincidently helped to build up blocks, which is a key finding and was applied in this work to synthesise cross-linked PHA-*b*-PHA block copolymers that might potentially improve the toughness of PHAs.

Based on the above discussion, the one-step synthesis was shown to be feasible for building molecular weight, with evidence of this being primarily through the formation of urethane groups, and the synthesis could be manipulated by extending the reaction time. In support of this, the molecular weights of the reaction products from the one-step synthesis in Experiment #2B almost trebled over the initial 18 h, with $M_{n}$ increasing from 24 to 67 kDa and $M_{w}$ increasing from 43 to 126 kDa (Figure 2b). As the reaction continued to progress, the $M_{n}$ and $M_{n}$ of the final reaction products (after 24 h) increased around fivefold overall compared to the starting hydroxy-functionalised PHBV. Once again, it is likely that the formation of allophanate linkages—causing branching and chemical cross-linking—contributed greatly to the increase in molecular weight (with 19.5 wt.% insoluble gel, Table 2). It is therefore assumed that the synthesised final product—where the molecular weight increased fivefold—is likely a mixture of linear, branched and/or cross-linked block copolymers. The ^1^H-NMR spectra were very similar to those of Experiment #2A, and, hence, are not discussed in detail here. 

### 3.2. Thermal Properties of As-Synthesised Block Copolymers

The thermal properties of the as-synthesised final products are given in Table 3. The respective DSC thermograms are shown in the Appendix A. Three random copolymers of 1 mol% 3HV (Random_1HV#1, Random_1HV#2 and Random_1HV#3) were compared with the two block copolymer products from Experiment #2A (Block_1HV#1) and #2B (Block_1HV#2). The product from Experiment #1 was not included as it was a mixture of block copolymers and the unreacted starting random copolymers. Comparing the initial cell-produced random copolymer of 1 mol% 3HV (Random_1HV#3, $M_{n}\;$of 194 kDa) with the hydroxy-functionalised transesterification products (Random_1HV#1, $M_{n}\;$of 86 kDa and Random_1HV#2, $M_{n}\;$of 117 kDa), the thermal properties of the PHBV random copolymers were similar, except for the slight increase in $T_{g}$ for Random_1HV#3, which was due to the lower chain mobility of the longer polymer chains. 

However, the thermal properties differed slightly more between random copolymers and the chemically synthesised block copolymers, with slightly lower $T_{m}$ and $∆H_{m}$ values and a small increase in the $T_{g}$ being observed for the latter. This is likely due to constrained chain mobility as a consequence of the formation of three-dimensional allophanate or potential biuret structures [53], as well as possible hydrogen bonding effects from the urethane groups (although they are a small component of the composition overall) and possible further constraints due to microphase separation between links. 

By comparing the two synthesised block copolymers of different degrees of cross-linking (Block_1HV#1 and Block_1HV#2), it is also observed that with the increase in cross-linking, the $T_{m}$ and $∆H_{m}$ values decreased slightly, while the $T_{g}$ values increased from 6.3 °C (for Block_1HV#1) to 8.9 °C (for Block_1HV#2). 

## 4. Conclusions and Outlook

Overall, a comparison between the one-step and the two-step synthesis of PHBV-*b*-PHBV block copolymers has been established, based on relatively high molecular weight hydroxy-functionalised copolymeric PHBV starting materials containing 1 mol% 3HV. In both strategies, block copolymers were successfully synthesised, as evidenced by the increase in molecular weight and the formation of urethane groups from NMR analysis. However, the one-step synthesis is a more efficient approach. Moreover, there was evidence of by-product formation, particularly allophonate and likely biuret groups, through the further reaction of isocyanates with the urethane groups initially formed. This resulted in cross-linking, particularly following extended reactions with high isocyanate loadings, and the formation of relatively high molecular weight products. It is a key finding of this work and helps to establish an approach for the synthesis of novel PHA-*b*-PHA through the relatively simple one-step strategy of block copolymer production. In addition, the fundamentals of the synthesis were investigated, which, to our knowledge, have not been reported for the synthesis of PHA-*b*-PHA block copolymers based on diisocyanate chemistry.

It is also suggested for future research that the effect of reaction conditions—such as temperature and amounts of catalyst—on the formation of the side reactions of the diisocyanate chemistry could be investigated. By reducing or even eliminating the side reactions—i.e., maximising the formation of isocyanate-terminated intermediate reaction products based on a [NCO]/[OH] ratio of 2.2—the synthesis of a target PHA-*b*-PHA-*b*-PHA tri-block copolymers would be achievable.

## Figures and Tables

**Figure 1 polymers-15-03257-f001:**
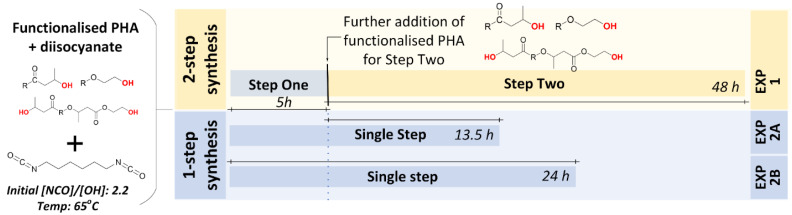
Graphical overview of PHBV-*b*-PHBV block copolymer syntheses, where R represents PHBV main chain. The yellow colour indicates the 2-step synthesis while the blue colour indicates the one-step synthesis.

**Figure 2 polymers-15-03257-f002:**
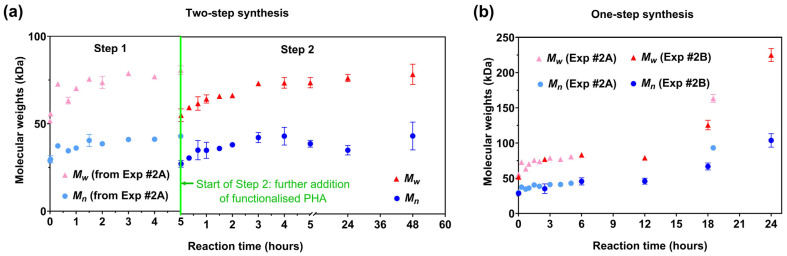
Molecular weights of the soluble fraction of reaction products in (**a**) a two-step block copolymer synthesis (Experiment #1); (**b**) a one-step synthesis of block copolymer (Experiment #2A and 2B). Values are shown as mean with standard deviation; the error bar may not be visible due to the marker. Note: different scales for molecular weight in (**a**,**b**).

**Figure 3 polymers-15-03257-f003:**
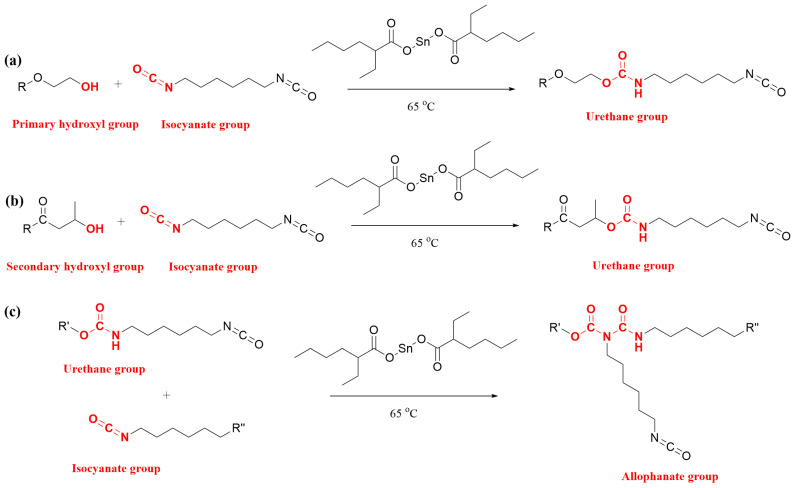
Chemical reaction of (**a**) primary hydroxyl groups with isocyanate groups to form urethane groups; (**b**) secondary hydroxyl groups with isocyanate groups to form urethane groups; and (**c**) urethane groups with isocyanate groups to form allophanate groups. R and R′ = PHBV random copolymer and R″ = NCO or NHC(O)OR′. The functional groups of interest are identified in red.

**Figure 4 polymers-15-03257-f004:**
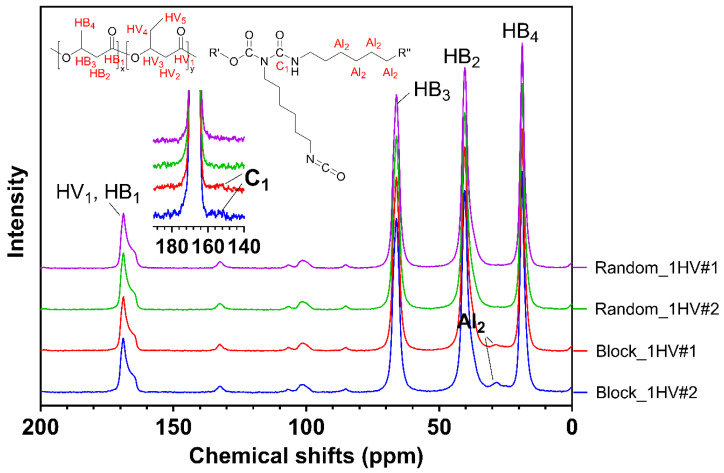
Solid-state ^13^C-NMR results of PHBV materials of 1 mol% 3HV, where R′ = PHBV random copolymer and R″ = NCO or NHC(O)OR′. The carbons identified in the spectra are indicated in red in the chemical structures above.

**Figure 5 polymers-15-03257-f005:**
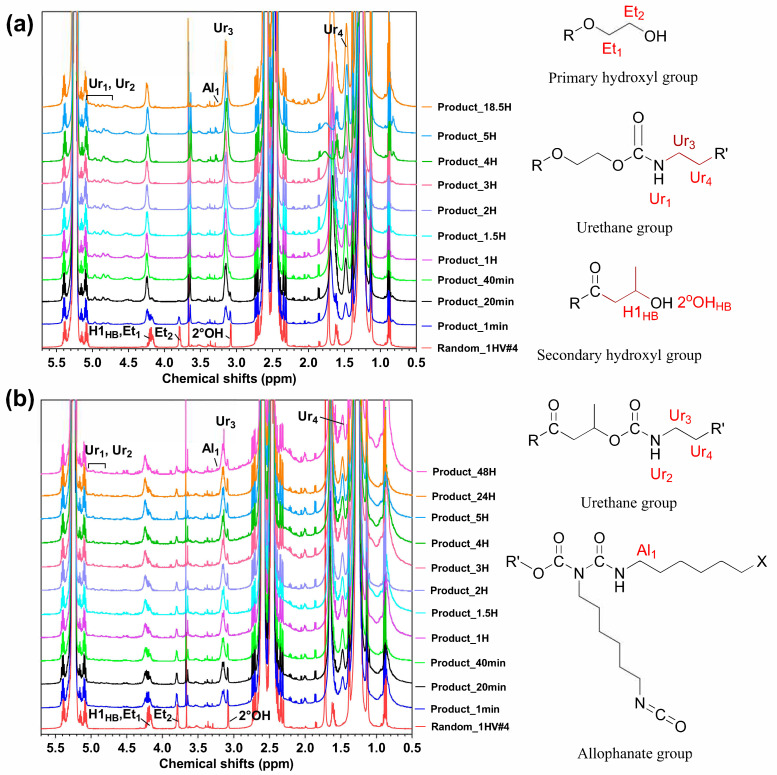
Detailed ^1^H-NMR results of the reaction products of (**a**) the first step of the two-step synthesis (from the first 5 h of Experiment #2A) and (**b**) the second step of the two-step synthesis (Experiment #1), where time is post addition of second tranche of hydroxy-functionalised PHBV. Key functional groups identified in the ^1^H-NMR spectra through labels are shown to the right, labelled in red. Et = ethylene glycol-derived end group, where Et_1_ and Et_2_ are the protons associated with that ethylene group; H1_,HB_ is the tertiary proton associated with the butyric acid derived end group; 2 °OH = secondary hydroxyl proton associated with the butyric acid derived end group; Ur_1_ = proton of primary hydroxyl-derived urethane group; Ur_2_ = proton of secondary hydroxyl-derived urethane group; Ur_3_ and Ur_4_ = protons adjacent to the urethane groups that were derived from the diisocyanate; Al_1_ = proton adjacent to allophanate group; R and R′ = PHBV random copolymer; and X = NCO or NHC(O)OR′.

**Table 1 polymers-15-03257-t001:** Details of reaction conditions and materials used in block copolymer syntheses based on 1 mol% 3HV content PHBV.

Exp’t No.	Hydroxy-functionalised PHBV Starting Materials	$\bar{M_{n}}$ (kDa)	$\bar{M_{w}}$ (kDa)	*Đ*	3HV (mol%)	1 °OH and 2 °OH Used in Reaction (mmoles/ g PHA)	Description	Experimental Conditions
1	Random_1HV#4	28	51	1.8	1	0.084	First and second steps of the two-step synthesis	[NCO]/[OH] for step 1 = 2.2; Temp = 65 °C; Time (step 1) = 5 h; Theoretical [NCO]/[OH] for step 2 = 0.5 *; Temp = 65 °C; Time (step 2) = 48 h*Overall [NCO]/[OH] = 2.2:3*
2A	Random_1HV#4	28	51	1.8	1	0.084	One-step synthesis (first step of the two-step synthesis)	[NCO]/[OH] for step 1 = 2.2; Temp = 65 °C: Time = 18.5 h
2B	Random_1HV#5	25	44	1.7	1	0.103	One-step synthesis extended	[NCO]/[OH] = 2.2; Temp = 65 °C: Time = 24 h

* Based on initial NCO addition in step 1.

**Table 2 polymers-15-03257-t002:** Solubility of final reaction products of block copolymer synthesis.

Experiment No.	Final Reaction Products	Insoluble Component (wt.%)
1	Product of two-step synthesis	2.7
2A	Block_1HV#1	4.4
2B	Block_1HV#2	19.5

**Table 3 polymers-15-03257-t003:** Thermal properties of PHBV-*b*-PHBV block copolymers and random copolymers.

	PHBV Material	$M_{n}$ (kDa)	$M_{w}$ (kDa)	*Đ*	$T_{m}$ (°C)	$∆H_{m}$ (J/g)	$T_{c}$ (°C)	$∆H_{c}$ (J/g)	$T_{g}$ (°C)
random	Random_1HV#1	86	186	2.2	157.6/ 174.0	102.3	98.3	87.1	2.6
	Random_1HV#2	117	273	2.3	160.0/ 175.0	102.5	103.4	89.2	2.6
	Random_1HV#3	194	455	2.3	159.5/ 170.0	106.0	104.7	87.3	4.0
block	Block_1HV#1	94	159	1.7	153.1/ 168.5	94.0	104.9	78.9	6.3
	Block_1HV#2	107	221	2.1	152.2/ 167.3	89.2	110.0	83.7	8.9

## Data Availability

The raw/processed data required to reproduce these findings cannot be shared at this time due to technical or time limitations.

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
