# Peer review of "Synthesis and Characterisation of Poly(3-hydroxybutyrate-co-3-hydroxyvalerate)-b-poly(3-hydroxybutyrate-co-3-hydroxyvalerate) Multi-Block Copolymers Produced Using Diisocyanate Chemistry"

_polymers, 2023, doi:10.3390/polym15153257_

Round 1
Reviewer 1 Report
In this work, Mai et.al synthesized PHBV-b-PHBV block copolymers based on high molecular weight hydroxy-functionalized PHBV connected by diisocyanate chemistry. The synthesis method including two-step and on-step methods were compared and both methods can form multiblock architecture. At the same time, one-step method is helpful to build up molecular weight. Compared with random copolymer species, block-copolymer type PHBV materials show good thermal properties with decreased Tm when having associated branches or crosslinking point. Before final acceptance, a revision must be conducted to improve the readability and rationality.
1. About the Abstract, the author should point out the potential technical or scientific problem that encountered, further highlight the key findings and novelty of this work. Some expressions should be modified.
2. For the introduction part, the logical flow and structure can be improved into four main parts. The first part introduced PHA, the importance of copolymer synthesis and potential applications. The second part can focus on the previous reported methods to prepare PHA random copolymers and block copolymers. The advantages and difficulty existed should be pointed out. The third part can further highlight the principle, advantages of isocyanate chemistry. At last, to give readers a short impression on novelty of this work and the main research experiments related with the topic.
3. Figure 3a present Mw of product during two-step process. The green line split two individual steps. So why the Mw at this line is not the same value? The Y scale bar should be zoomed to highlight the evolution process or Mw different at different reaction time.
4. Table 2 shows solubility of different samples. what is the gel part of byproduct possibly be in these samples?
5. Figure 5 is NMR data. X axis indicates wavenumber? Please double check the plot.
6. Table 3 shows thermal properties of these copolymers. How about the physical properties of these samples like mechanical property since the author discussed the synthesis of block copolymer can enhance physical properties. The author should check this hypothesis.
English quanlity is enough.
Author Response
Please see the response in the attached document.

Reviewer 2 Report
The authors described a nice approach for the synthesis of PHA-based block copolymer via diisocyanate initiation. PHAs are attractive bio-based new plastics and developing new polymerization methods to make block-copolymers is attractive. Although side reactions might be present and further studies are required, this work deserves publication in Polymers.
1. The authors claim that the thermal properties of block-PHAs are different than random-PHAs. The reviewers, however, find only very limited difference between their thermal properties.
2. Can the authors comment on the relatively large PDIs (~2)?
3. These polymers seem to have lower solubilities in CHCl3, since heating at 80C is necessary to dissolve them. Maybe THF based GPCs are more suitable for these polymers?
Reviewer 3 Report
This paper suggests that the synthesis methods of the PHBV-b-PHBV block copolymers using PHA macroinitiators with relatively high moleuclar weight. However, it is difficult to understand what the motivation of this methods used in this paper is. The part of the description seems to be insufficient for the readers' understandings.
- Around the ends of the Introduction, the readers would understand the importance of the synthesis of the high molecular weight PHBV-b-PHBV block copolymers. However it is difficult to speculate the motivation of the synthesis method that the author used. It is better to mention the clear explanaton of the reason why the author used the synthtesis methods. (Or the advantaged of the methods that the author used should be clearly metioned.)
- The paragraph starts at line 123 on page 3, it is not easy to understand. For figure 3a, the readears would feel wonder why the molecular weights are decreased after the start of Step 2.
- In Introduction, the mechanical properties of the block copolymer are mentioned. However, there are no data of mechanical properties of the block copolymer that the author synthesized. It would be better to add the data of the mechanical performances of them. The Introduction seems to be improper quantities. Only related infromation should be supplied to the readers.
Round 2
Reviewer 3 Report
The aim and significance of this study would be understandable for the readers. I agree to be published.